# Evolutionary and Structural Analysis of PP16 in Viridiplantae

**DOI:** 10.3390/ijms25052839

**Published:** 2024-02-29

**Authors:** Domingo Jiménez-López, Beatriz Xoconostle-Cázares, Berenice Calderón-Pérez, Brenda Yazmín Vargas-Hernández, Leandro Alberto Núñez-Muñoz, José Abrahán Ramírez-Pool, Roberto Ruiz-Medrano

**Affiliations:** Departamento de Biotecnología y Bioingeniería, Centro de Investigación y de Estudios Avanzados, Av. Instituto Politécnico Nacional 2508, Mexico City 07360, Mexicobcalderon@cinvestav.mx (B.C.-P.); leandro.nunez@cinvestav.mx (L.A.N.-M.); jramirezp@cinvestav.mx (J.A.R.-P.)

**Keywords:** phloem protein 16, phylogeny, phloem transport, non-cell autonomous protein

## Abstract

Members of the phloem protein 16 (PP16) gene family are induced by elicitors in rice and the corresponding proteins from cucurbits, which display RNA binding and intercellular transport activities, are accumulated in phloem sap. These proteins facilitate the movement of protein complexes through the phloem translocation flow and may be involved in the response to water deficit, among other functions. However, there is scant information regarding their function in other plants, including the identification of paralog genes in non-vascular plants and chlorophytes. In the present work, an evolutionary and structural analysis of the PP16 family in green plants (Viridiplantae) was carried out. Data mining in different databases indicated that *PP16* likely originated from a larger gene present in an ancestral lineage that gave rise to chlorophytes and multicellular plants. This gene encodes a protein related to synaptotagmin, which is involved in vesicular transport in animal systems, although other members of this family play a role in lipid turnover in endomembranes and organelles. These proteins contain a membrane-binding C2 domain shared with PP16 proteins in vascular plants. In silico analysis of the predicted structure of the PP16 protein family identified several β-sheets, one α-helix, and intrinsically disordered regions. PP16 may have been originally involved in vesicular trafficking and/or membrane maintenance but specialized in long-distance signaling during the emergence of the plant vascular system.

## 1. Introduction

The vascular system of higher plants consists of two conducting tissues: xylem and phloem. Xylem vessels are responsible for the transport and distribution of water and minerals from the roots to the aerial parts of the plant. On the other hand, the phloem is involved in the transport and distribution of photosynthates from source to sink tissues, as well as long-distance signaling molecules that coordinate plant development and responses to external cues [1]. In angiosperms, the functional phloem is composed of enucleated sieve elements (SEs) and nucleated companion cells (CCs) interconnected through specialized branched plasmodesmata (PDs). PDs facilitate the delivery of nutrients and signaling molecules into the long-distance translocation stream [2]. Although several examples involving long-distance signaling in plants based on proteins and RNA have been described (e.g., flower induction, systemic acquired resistance, tuberization, and phosphate allocation) [1,2,3], supracellular control mechanisms are largely unknown.

The specific mechanisms and proteins involved in mesophyll PD size exclusion limit (SEL) changes can vary depending on the physiological context, developmental stage, and environmental conditions [4,5]. However, the PD at the boundary of SE-CC has an SEL of around 40 kDa. Consequently, smaller proteins would freely diffuse between these two cell types [1,6,7]. Some SE proteins are involved in sieve tube maintenance and function, while the function of others either implies a signaling function or has not been characterized to date [8].

Cucurbits are the most extensively studied plant family concerning proteins and RNA in the phloem. Analytical quantities of phloem and xylem sap exudates can be obtained from cucurbit plants, but recent methods have also enabled the collection of phloem-enriched exudates from species such as tomato, Arabidopsis and Citrus [9,10,11]. The cucurbit phloem proteome contains several RNA-binding proteins (RBPs) consistent with the presence of many RNA species including mRNAs, miRNAs, and lncRNAs [8,12,13]. Indeed, one of the most abundant proteins in the pumpkin phloem translocation stream is the *Cucurbita maxima* 16 kDa phloem protein (CmPP16). CmPP16 exhibits sequence-independent RNA-binding activity and is able to translocate cell to cell and to distant tissues [14]. Furthermore, CmPP16 forms stable protein–protein complexes within the phloem sap, likely traveling as heterocomplexes [15,16]. Overexpression of *CmPP16* in pumpkins induces drought tolerance, thus suggesting its role in responses to stress through long-distance signaling [17]. On the other hand, *PP16* transcripts are induced by elicitors in rice [18], supporting the notion that members of this gene family are involved in response to biotic and abiotic stress.

Similarly, *Citrus sinensis* (sweet orange) phloem protein 16-1 (CsPP16-1) enables the transport of translationally fused antimicrobial peptides into the phloem translocation stream, thus mitigating citrus greening disease (Huanglongbing) associated with *Candidatus* Liberibacter asiaticus, which is restricted to the plant vasculature [19,20]. However, its precise mechanism of action remains unknown. The fact that two plant species (*C. maxima* and *C. sinensis*) with different modes of phloem loading and unloading appear to harbor PP16 homologs that enter the long-distance phloem translocation stream suggests that their function is conserved, at least in vascular plants.

Previous phylogenetic analysis suggested that this gene is widespread in land plants, including tracheophytes and bryophytes. However, chlorophytes possess a larger version of this gene. The *PP16* gene family shares a lipid-binding C2 domain with its chlorophyte homologs. Presumably, a duplication event in more recent lineages gave rise to this gene [18]. To gain further insight into its origin and function, phylogenetic and structural analyses were performed. The results indicated that the chlorophyte proteins related to PP16 are an independent lineage from all extant multicellular plants and do not harbor any actual PP16 homologs. However, all these proteins are related to the extended synaptotagmin proteins (E-SYTs), which participate in membrane maintenance and lipid turnover in endomembranes and organelles [21]. We propose that there are two PP16 lineages, referred to as A and B, based on the presence of two insertions and conserved divergent motifs. Finally, protein structure prediction suggested the structural and functional conservation of PP16 among vascular plants.

## 2. Results

### 2.1. Identification of PP16 Proteins in Viridiplantae

PP16 homologs were identified in the extant plant genome databases using CmPP16 from *C. maxima* as a query. The plant species selected for analysis were as follows: six green algae (chlorophytes), two basal embryophytes, one basal angiosperm, five monocots, and twenty dicots, with the PP16 homologs for each species (Appendix A). No clear PP16 orthologs were found in chlorophytes (unicellular green algae) since the global identity was low (<32%) with the query sequence. Although local sequence alignment of CmPP16- and PP16-related proteins in green algae was performed (Appendix A), these sequences were excluded from the multiple sequence alignment for evolutionary analysis. Two species of basal embryophytes (*Physcomitrium patens* and *Selaginella moellendorffii*) harbored proteins with ≈33% identity to PP16. In the basal angiosperm *Amborella trichopoda*, two PP16 orthologs were found, while the five species of monocots used in the analysis comprised three to four PP16 orthologs, except for *Panicum virgatum,* which harbored seven PP16 orthologs. The twenty dicot genomes species analyzed in this study comprised two to three PP16 orthologs, except for *Phaseolus vulgaris*, *Glycine max*, *Prunus persica, Cucurbita moschata,* and *Cucurbita maxima*, which contained six, eleven, seven, six, and five PP16 orthologs, respectively (Figure 1).

### 2.2. Phylogenetic Distribution of PP16 Proteins in the Viridiplantae Family

To determine the evolutionary history of PP16 protein family members, phylogenies were analyzed based on the full-length sequences of 104 PP16 proteins in 28 Viridiplantae species. The phylogenies were calculated using ML and NJ methods. Two distinct clades were obtained with both phylogenetic analyses, which were named group A and group B (Figure 2). Group B was supported with a 59% bootstrap value using the NJ method and 56% using the ML method. *P. patens* and *S. moellendorffii* were positioned as basal embryophytes (orange branch) and used as outgroups. Two proteins from *A. trichopoda* (atr|Amborella_trichopoda_v1.0_scaffold00009.330 and atr|Amborella_trichopoda_v1.0_scaffold00009.331) were positioned in group A in both trees (yellow branch). *C. maxima* harbored five PP16 paralogs, four (cmax|Cucurbita_maxima_Q9ZT47.3_PP16-1, cmax|Cucurbita_maxima_AAY96411.1_PP16-2, cmax|Cucurbita_maxima_XP_022991040.1_PP16-1-like_isoform-X1, and cmax|Cucurbita_maxima_XP_022992698.1_ERG1-like) fell into group B (black color asterisk) and one (cmax|Cucurbita_maxima_XP_022996373.1_ERG3-like) into group A (black color asterisk, Figure 2), regardless of the method employed. The *C. sinensis* genome contains three PP16 homologs, two positioned in group A and one in group B (blue color asterisk, Figure 2).

To unveil the differences between groups A and B, the multiple sequence alignment of the 104 PP16 ortholog sequences was analyzed. The analysis indicated that in group B, there were two insertions marked as red bars (Figure 3). Insertion 1 consists of four residues (consensus sequence: PGSX) and insertion 2 of six residues (consensus sequence: VE(N/K)G(X)(A/S/Y).

### 2.3. Ancestral State Reconstruction of PP16 Orthologs

On the other hand, to elucidate whether PP16 in Viridiplantae has a common origin, ancestral reconstruction was performed. Three main clades were obtained: basal embryophytes, group A, and group B (red, yellow, and green color, respectively; Figure 4), which is consistent with phylogenetic analyses carried out by the ML and NJ methods. The basal embryophytes clade includes *P. patens* and *S. moellendorffii* orthologs. Group A is composed of fifty-two PP16 orthologs, which include one of the five CmPP16 orthologs (cmax_Cucurbita_maxima_XP_022996373_1_ERG3_like). Group B is composed of forty-nine PP16 orthologs and includes four of the five CmPP16 orthologs (cmax_Cucurbita_maxima_XP_022992698_1_ERG1_like, cmax_Cucurbita_maxima_XP_022991040_1_PP16_1_like_isoform_X1, cmax_Cucurbita_maxima_AAY96411_1_PP16_2 and cmax_Cucurbita_maxima_Q9ZT47_3_PP16_1). It is possible that one of the orthologs of the basal embryophyte clade (first ancestral node) gave rise to group A (second ancestral node) and eventually to group B (third ancestral node).

### 2.4. Overall Domain Architecture of PP16 Proteins

To determine the overall similarity as well as a motif/domain architecture of the PP16 proteins, common motifs were analyzed. First, potential common motifs were identified in the C2 domain using 104 sequences of the PP16 proteins. Second, motifs and domains found adjacent to the C2 domain were also searched. A consensus, represented as the logo of the C2 domain, has already been reported previously [14] and is available in the Kyoto Encyclopedia of Genes and Genomes (KEGG) database (https://www.genome.jp/tools/motif/; accessed on 22 November 2023). A list of the C2 domains present in each of the 104 sequences of PP16 peptides is shown in Appendix A. Two non-redundant motifs were obtained in the C-terminus region of the proteins (motif 1 and motif 2; Figure 5). Likewise, the analysis of the domain architecture of the A and B groups resulted in only one common motif within the N-terminal of the C2 domain and two common motifs at the carboxy end. Despite the fact that the motifs in both groups displayed high similarity, there were some differences. There are two insertions (insertion 1 and insertion 2, black rectangle; Figure 5) in group B. Insertion 1 is in the C2 domain and insertion 2 is located adjacent to motif 1.

### 2.5. Structural Analysis of the PP16 Proteins by Template-Based Modelling

The crystal structure of most PP16 proteins has not been elucidated. Nevertheless, the 3D structure of the phloem protein 16-1 of *A. thaliana* (AtPP16-1) has been obtained through nuclear magnetic resonance (NMR) spectroscopy [22]. Thus, AtPP16-1 was used as a template model to predict the structure of PP16 homologs with the AlphaFold2 server. AtPP16-1 consists mostly of random coil-like disordered regions and overhanging side chains, except for three β-sheets and one α-helix. In addition, the N-terminal disordered regions of AtPP16-1 form a small lobe, which connects to the remaining protein region through an irregularly shaped cleft [22]. The results of this analysis showed that predicted structures of CmPP16-1, CmPP16-2, CmPP16-1-like_isoform-X1, Cm_XP_022992698.1_ERG1-like, and CsPP16-1 (group B) were composed of nine β-sheets and one α-helix (Figure 6). Similarly, Cm_XP_022996373.1, CsPP16-2, and CsPP16-3 (group A) predicted structures were composed of eight β-sheets and one α-helix shorter than the structures of group B (Figure 6).

### 2.6. PP16 Proteins Are Related to the Extended Synaptotagmin (E-SYT) Protein Family

A more comprehensive analysis of the PP16 sequence and its closest counterpart in chlorophytes was conducted. A BLASTp was performed using the CmPP16 amino acid sequence as a query against the corresponding protein sequences in chlorophyte species reported in the Phytozome v13 database. While all the chlorophyte proteins showed similarity to synaptotagmins, which are involved in tethering opposing membranes for vesicular fusion [23], not all of them appeared to be functional homologs. Furthermore, the localization of the C2 domain within the chlorophyte sequences varied widely. In some cases, it was found in the N domain, while in others, it was located at the C-end or in the central region (Figure 7).

More importantly, the most significant (lowest) e-value observed after aligning PP16 with its chlorophyte counterparts was found in *C. reinhardtii* (2.58 × 10^−9^). However, similar values, generally not highly significant, were also found with other species. The similarity was restricted to the shared C2 domain. Additionally, the size of the chlorophyte PP16 counterparts varied between 268 and 2253 amino acids. In the latter case, in *Botryococcus braunii*, the only similarity was the C2 domain, and this protein is annotated as related to chloroplast movement (by homology), so its relationship with other synaptotagmins is not clear.

Furthermore, although the PP16 homologs in chlorophytes are poorly characterized, they appear to belong to the extended synaptotagmin (E-SYT) family, which are resident proteins found in the endoplasmic reticulum. E-SYTs are involved in membrane contact rather than fusion and facilitate lipid exchange between membranes [21]. The embryophyte synaptotagmins more closely related to PP16 show a similar function [24]. Conversely, chlorophyte (*C. reinhardtii* and *V. carteri*) synaptotagmins showed similarity restricted to the C2 domain of E-SYTs of similar sizes (260–280 amino acids) when compared to embryophyte sequences, although more frequently to PP16 homologs in this lineage (not shown). Indeed, the Cre01.g015500_4532.1 transcript, which is the most similar in terms of e-value to CmPP16, encoded an E-SYT-like protein (by homology). Its closest homolog from *Lotus japonicum* is also a synaptotagmin-like protein.

However, PP16 proteins showed similar e-values to other embryophytes, both monocots and dicots, in which the similarity was also restricted to the C2 domain. Interestingly, several sequences with a similar e-value were found in bryophytes (*Marchantia polymorpha*, *Physcomitrium patens*, *Ceratodon purpureus*, and *Sphagnum phallax*), but corresponded to PP16 homologs as well as extended synaptotagmins, and synaptotagmin-related proteins harboring additional domains, such as an ADP-ribosylation factor-like domain. These results support the notion that the *PP16* gene in embryophytes originated from an ancestral extended synaptotagmin gene in species belonging to the common lineage of chlorophyta and embryophyta, and this gene was maintained in the latter.

### 2.7. Analysis of Selective Pressure of PP16 Genes

To detect if positive selection of the PP16 gene had an impact on its evolution, an analysis of this dataset was performed using the branch-site unrestricted statistical test for episodic diversification (BUSTED) method in the Datamonkey server. The BUSTED method allows us to test whether a gene has experienced positive selection in at least one site per branch or in the entire phylogenetic tree. On the other hand, to detect positive selection in specific sites, the mixed-effects model of evolution (MEME) and the fast, unconstrained Bayesian approximation (FUBAR) methods of the Datamonkey server were used. The results of the BUSTED method indicated that based on the likelihood ratio test, there is evidence of episodic diversifying selection or positive diversifying selection in the dataset analyzed, with p equal to 0.000007451 (http://datamonkey.org/busted/652eac01353125059b663a27). On the other hand, the result of the MEME algorithm indicated that there are twelve sites of episodic positive/diversifying selection (6, 8, 58, 98, 99, 139, 175, 198, 205, 216, 218, and 237) with a *p*-value threshold of 0.05 (Figure 8, red color font, http://datamonkey.org/meme/652e7173353125059b6635a1). In addition, the results of the FUBAR analysis indicated that there is only 1 site of pervasive positive/diversifying selection (183) and 128 sites of pervasive negative/purifying selection with a posterior probability of 0.9 (Figure 8, blue color font, http://datamonkey.org/fubar/652e9572353125059b663956).

## 3. Discussion

CmPP16 is a 16 kDa RBP that accumulates in the angiosperm CC-SE complex [1,14]. Evidence suggests that it binds certain mRNAs preferentially in the CC and moves to the SE through the plasmodesmata, interconnecting both cell types, along with bound mRNA into the SE for phloem transport [14,15]. Experimental evidence supports diverse functions of PP16 in plants including cell-to-cell movement and non-specific RNA binding [14]. Some of these functions have been corroborated in *A. thaliana* [22]. Additionally, PP16 is involved in regulating root growth, resource allocation in meristems, and interaction with plasmodesmata, facilitating its own cellular transport and selectively directing endogenous RNAs to different plant tissues across taxa, such as cucurbits and rice, highlighting the conservation of its functions [25,26,27]. Moreover, PP16 acts as a redox sensor in the extra fascicular phloem of cucurbits and a wound-responsive gene. It is susceptible to nitric oxide (NO) modifications and is involved in stress-induced redox signaling [28]. These interactions with NO may explain drought-tolerant phenotypes such as increased photosynthetic rates in overexpressing plants under drought conditions [17]. Additionally, PP16 has recently been described as a valuable tool that can be translationally fused to antimicrobial peptides for selective targeting to the phloem, combating diseases within the vascular system [19,20].

The molecular mechanisms and regulatory factors governing PP16 expression remain poorly understood. Therefore, it is essential to investigate the expression of this gene under stress conditions and developmental stages. Understanding the regulation of PP16 expression can provide valuable insights into its potential role in different biological processes. Meanwhile, graft mobility assays in non-cucurbitaceous species can provide insights into the long-distance movement capabilities of PP16 across a broader evolutionary context, as well as heterologous expression of these genes in chlorophytes, which do not harbor them.

Nevertheless, there is scarce information on the evolutionary origin of PP16, as well as its functions in other vascular (for example, those in which phloem loading is apoplastic rather than symplasmic) and non-vascular plants. Few studies have focused on PP16 using bioinformatics tools [14,18,29]. In this study, phylogenetic and structural analyses of PP16 homologs in different plant lineages were carried out to gain a better understanding of their evolution and functions. Identification of PP16 orthologs using BLAST in the Phytozome v13 database indicated that there were at least two homologs in different taxa (two in basal embryophytes, two in basal angiosperms, three to four in monocots, and two to three in dicots) in 28 species. The elevated number of PP16 orthologs in certain species such as *P. vulgaris*, *G. max*, *P. virgatum,* and *P. persica* (Figure 1) may have resulted from whole-genome duplication events, polyploidization, or interchromosomic rearrangements events [30,31,32,33]. In addition, PP16 orthologs are present only in embryophytes (specifically, those that harbor differentiated phloem tissue), such as *A. trichopoda*, monocots, and dicots.

Notably, a BLAST search for PP16-like DNA or protein sequences in gymnosperms yielded results only in *Ginkgo biloba*; however, E-SYT homologs were identified in the former species and in *Picea abies*. This could be due to incomplete coverage of these genomes.

A PP16 ortholog was likely established for the first time in a sister species of *A. trichopoda*, which is considered a basal species of flowering plants, once the vascular tissue evolved and eventually acquired the function of phloem RNA transport [14]. Nevertheless, the C2 domain of PP16 orthologs is present in most living systems, from eubacteria to eukarya [34]. Therefore, it is possible that *PP16* genes are derived from a larger gene in the common ancestor of chlorophytes and multicellular plants that includes a C2 domain. Indeed, the analysis presented herein supports this notion. The ancestral gene splits to give rise to *PP16* and another gene, the latter with a similar function to that of chlorophytes. The closest homologs of these proteins in embryophytes are E-SYT proteins of varying lengths but are usually smaller than E-SYT proteins from chlorophytes. The deletions/insertions, re-arrangement of exons/introns, or combinations of protein modules could explain the origin of the PP16 orthologs with the C2 domain [35]. Furthermore, PP16 harbors a C2 membrane-binding domain that may be important for its function.

The 104 PP16 peptide sequences used in this study (excluding protein sequences from six species of green algae) can be classified as A and B in a phylogenetic tree created by NJ and ML methods (Figure 2). By analyzing multiple sequence alignment, we suggested that group B originated from group A by a duplication event. Insertions 1 and 2 (Figure 3), localized between the C2 domain and motif 1 of group B of the PP16 orthologs, supported this notion. In addition, the PP16 homologs from basal embryophytes (*P. patens*, *S. moellendorffii*, and *A. trichopoda*; group A) (Figure 2 and Figure 3) lack these insertions, which further suggests that the orthologs from group B are a product of the duplication of members of group A. Similarly, this phylogenetic inference can be supported by the ancestry analysis (Figure 4). One of the orthologs of the basal embryophyte clade (first ancestral node) may have given origin to group A (second ancestral node) and eventually to group B (third ancestral node).

Analysis of the domain architecture of PP16 proteins by MEME and Jalview revealed basic features in both groups (A and B). Both contained a C2 domain at the C-end and two distinctive motifs (motif 1 and motif 2) at the N-terminus (Figure 5). The evidence suggests that insertion 1 of group B, specifically the serine residue, and the tyrosine residue adjacent to insertion 1 are critical for post-translational modifications such as glycosylation and phosphorylation, respectively. Indeed, these residues are fundamental for cell-to-cell movement and their interaction with *Nicotiana tabacum* non-cellular autonomous pathway protein 1 (Nt- NCAPP1) [16].

The 3D structure analysis across AlphaFold2 indicated that the *C. maxima* and *C. sinensis* models (CmPP16-1, CmPP16-2, CmPP16-1-like_isoform-X1, Cm_XP_022992698.1, and CsPP16-1 of group B; and Cm_XP_022996373.1, CsPP16-2, and CsPP16-3 of group A) differed from the AtPP16-1 template [22]. The protein models of the A (Cm_XP_022996373.1, CsPP16-2, and CsPP16-3) and B groups (CmPP16-1, CmPP16-2, CmPP16-1-like_isoform-X1, Cm_XP_022992698.1_ERG1-like, and CsPP16-) contained nine β-sheets and eight β-sheets (Figure 6), respectively, compared to the AtPP16-1 template, which contained three β-sheets. The structural complexity of AtPP16-1 and signal distortions due to high sensitivity to motion in the NMR technique may explain the observed differences between the template and our models [36]. Considering that the available structural information of PP16 was determined with limited spectral resolution (which can affect the precision of resonance assignment and the determination of secondary structures), characterizing PP16 with additional tools such as multidimensional NMR, high-resolution mass spectrometry, electron paramagnetic resonance spectroscopy, high-resolution Fourier-transform infrared spectroscopy, high-resolution Raman spectroscopy, and even cryo-electron microscopy could allow for a more reliable structure.

Furthermore, the PREDCITER server indicates that the intrinsically disordered regions of most of the group A and B models are contained, mainly, in the amino and carboxy termini, which is consistent with the results reported previously [22].

On the other hand, insertions 1 and 2 affected the α-helix size in the 3D structure of Cm_XP_022996373.1_ERG3-like, CsPP16-2, and CsPP16-3 models (group A), respectively. However, it has not been determined that these insertions, or in any case the deletions, affect PP16’s function. Nevertheless, subsequent analyses using nuclear magnetic resonance, X-ray crystallography, and molecular docking for some of the models obtained from *C. maxima* and *C. sinensis* will be necessary to elucidate their structural and functional features.

Overall, our results indicate that bona fide PP16 proteins are present only in land plants and likely arose from the C2 domain of an extended E-SYT protein harbored by a common ancestor of chlorophytes and embryophytes. Given that the latter lineage harbors similar E-SYT proteins (although shorter in length than in chlorophytes), it is possible that the emergence of PP16 genes occurred through partial duplication of this gene. Furthermore, these proteins have a specific role in vascular tissue, particularly in phloem signaling and in response to stress. Finally, the highly conserved predicted structure of PP16 suggests a conserved function in embryophytes.

Selective pressure analysis of PP16 genes with MEME indicated that two of the twelve sites of episodic positive/diversifying selection (6 and 175) and one site of pervasive positive/diversifying selection obtained with FUBAR (183) were false positives (Figure 8). This erroneous inference of the MEME and FUBAR methods can be explained by a misalignment-related error of the aligner CLUSTAL [37]. The remaining sites revealed by MEME (8, 58, 98, 99, 139, 198, 205, 216, 218, and 237) underwent episodic positive/diversifying selection. The duplication of PP16 genes with accumulated deleterious mutations and residues with evolutionary adaptive potential (specifically for sites 58, 98, 139, 198, 205, and 237) could explain the evidence of episodic positive/diversifying selection. Nevertheless, the results of the FUBAR analysis revealed 128 sites under pervasive negative/purifying selection (including positions 6, 8, 58, 99, 216, and 218 of the MEME method). The sign of pervasive negative/purifying selection suggests its functional importance, as well as conservation of the C2 domain, motif 1, and motif 2 in PP16 genes [38]. Position 78 corresponds to tyrosine (UAU/UAC), a residue critical for phosphorylation, as well as for interaction with *Nicotiana tabacum* non-cellular autonomous pathway protein 1 (Nt- NCAPP1) [16], which is one of the 128 sites under pervasive negative/purifying selection. In fact, investigating the evolutionary origin of the C2 domain of PP16, considering its presence in other taxa, would be useful for understanding the functions of the domain in a broader taxonomic context.

## 4. Materials and Methods

### 4.1. Identification and Sequence Retrieval of PP16 Proteins in Viridiplantae Species

To analyze the phylogeny of members of the PP16 protein family in Viridiplantae, homologous amino acid sequences were retrieved from the Phytozome version 13 database (https://phytozome-next.jgi.doe.gov/) by a BLAST search using the CmPP16-1 protein sequence (GenBank accession number: Q9ZT47.3) from *C. maxima* as a query. Additionally, to retrieve sequences that were more distantly related to CmPP16-1 orthologs, CmPP16-2 (AAY96411.1) and CmPP16-1-like_isoform-X1 (XP_022991040.1) were also used as queries in the Phytozome database.

### 4.2. Sequence Alignment and Phylogenetic Analysis

PP16 protein sequences were aligned using Clustal Omega version 1.2.3 in local mode [39]. Jalview software version 1.0 [40] was used to analyze and edit the sequence alignments. The PP16 peptide sequence alignment was used for phylogenetic tree estimations using MEGAX version 10.1.8 software [41]. Phylogenetic analyses were made with 1000 bootstrap replicates [42] using the LG + gamma distributed with invariant sites (G+I) model for the maximum likelihood (ML) method and the Jones–Taylor–Thornton model for the neighbor-joining (NJ) method with 5 gamma categories and the complete deletion option. The two phylogenies were presented and annotated using EvolView version 3 (https://www.evolgenius.info/evolview). Phylogenies for conventional taxonomic classifications were obtained from the National Center of Biotechnology Information (NCBI) taxonomy server (http://www.ncbi.nlm.nih.gov/Taxonomy).

### 4.3. Ancestral State Reconstruction of PP16 Orthologs

For the ancestral PP16 reconstructions, PP16 ortholog amino acid sequences from Viridiplantae species were used. Phylogenetic trees were produced using Bayesian statistics (https://nbisweden.github.io/MrBayes/download.html). For this tree, 10 million generations were used to build the phylogeny, with parameter chains = 4, printfreq = 1000, samplefreq = 100, and burnin = 200. The tree was visualized in Mesquite software (https://www.mesquiteproject.org/), and the reconstruction of ancestral characters was carried out using the maximum parsimony algorithm.

### 4.4. Determination of Motifs and Generation of Logos

The Multiple Expectation maximization for Motif Elicitation (MEME) method using MEME Suite software (University of Queensland, St. Lucia, Australia) version 4.12.0 (http://meme-suite.org/) in local mode was used to search for conserved motifs within the PP16 protein sequences. To determine the C2 domain, values of zero or one site per sequence with 70 and 110 amino acids were used as minimum and maximum motif sizes, respectively. To determine motifs adjacent to the C2 domain, values of zero or one site per sequence with 6 and 110 amino acids were used as minimum and maximum motif sizes, respectively. In both cases, an E-value < e^−10^ was used as the cutoff. Jalview software version 1.0 was used to check and obtain logos for domains/motifs within the A and B groups.

### 4.5. Analysis of the Tridimensional Structure of PP16 Proteins by Template-Based Modelling

The AlphaFold2 server (https://colab.research.google.com/github/sokrypton/ColabFold/blob/main/AlphaFold2.ipynb) was used to determine the tridimensional (3D) structure of the following PP16 proteins: CmPP16-1, CmPP16-2, CmPP16-1-like_isoform-X1, Cm_XP_022992698.1_ERG1-like (XP_022992698.1) and Cm_XP_022996373.1_ERG3-like (XP_022996373.1) from *C. maxima*, and CsPP16-1 (XP_006486477.1), CsPP16-2 (XP_006477594.2) and CsPP16-3 (XP_006491029.1) from *C. sinensis*. The *Arabidopsis thaliana* PP16-1 (Protein Data Bank accession ID 5yq3) was used as a template model. The structure refinement of the PP16 protein models was carried out by GalaxyWEB (https://galaxy.seoklab.org/). UCSF CHIMERA software version 1.17.3 (https://www.cgl.ucsf.edu/chimera/) was used to analyze the 3D structures of the PP16 models. In addition, the intrinsically disordered regions of the PP16 protein models were predicted by using the PREDICTER server (http://original.disprot.org/pondr-fit.php).

### 4.6. Analysis of Selective Pressure of PP16 Genes

To analyze the selective pressure of the 104 PP16 genes in Viridiplantae, nucleotide sequences were retrieved from the Phytozome version 13 database (https://phytozome-next.jgi.doe.gov/) by a BLAST search using the CmPP16-1 protein sequence (GenBank accession number: Q9ZT47.3) from *C. maxima* as a query. Posteriorly, PP16 nucleotide sequences (stop codons previously removed) were aligned using CLUSTAL of MEGA version 11.0.11 software [41]. Finally, PP16 nucleotide sequence alignment was used to analyze the positive selection by the BUSTED, MEME, and FUBAR methods of the Datamonkey server (http://datamonkey.org/). The GARD (A Genetic Algorithm for Recombination Detection) method of the Datamonkey server (http://datamonkey.org/) was previously used to analyze recombination.

## Figures and Tables

**Figure 1 ijms-25-02839-f001:**
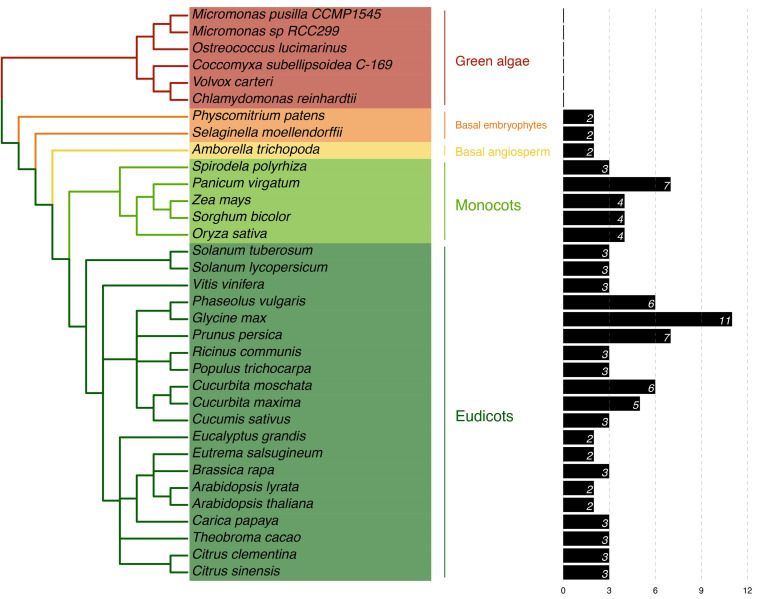
Phylogeny of Viridiplantae in relation to the presence of PP16 homologs. This analysis was carried out based on the *C. maxima* CmPP16 protein. To the right, the number of PP16 homologs in each plant species is shown.

**Figure 2 ijms-25-02839-f002:**
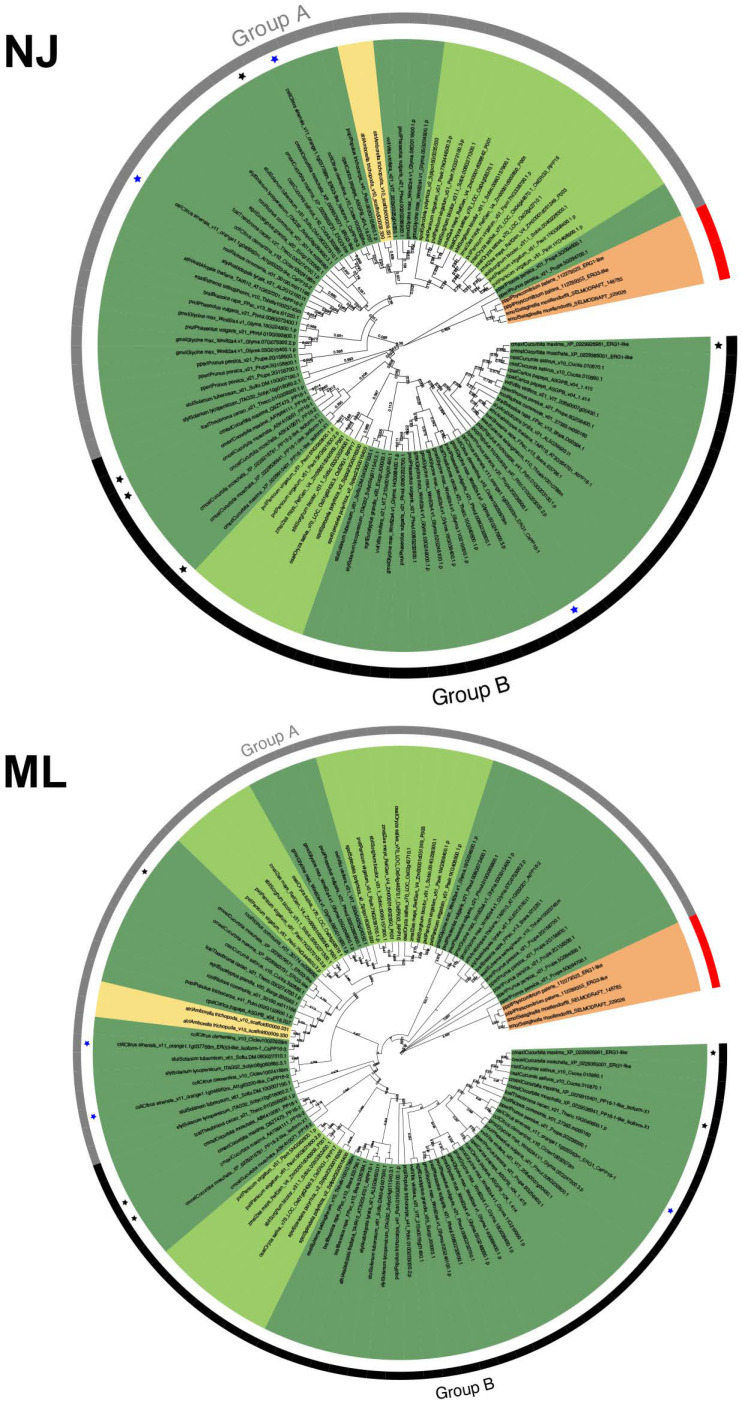
Phylogenetic analysis of the PP16 protein family in Viridiplantae. NJ shows the result of the phylogenetic analysis using the neighbor-joining method; ML is the result of the phylogenetic analysis using the maximum likelihood method. A bootstrap of 1000 was used. Both analyses yielded comparable results. Groups A and B in both trees are displayed as gray and black color strips, respectively. The black and blue asterisks indicate PP16 of *Cucurbita maxima* (CmPP16) and *Citrus sinensis* (CsPP16), respectively. Basal angiosperms are depicted in yellow, bryophytes in orange, monocots in light green, and eudicots in dark green.

**Figure 3 ijms-25-02839-f003:**
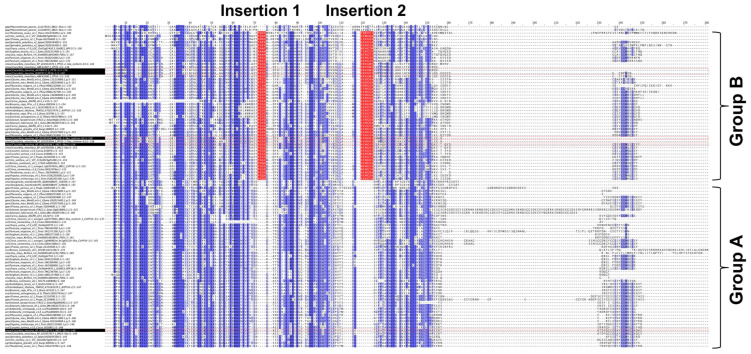
Multiple sequence alignment of 104 PP16 proteins from diverse embryophyte taxa. Dark blue highlights the identities between these proteins, and light blue the similarities. Groups A and B are displayed in the alignment. Insertions 1 and 2 of group B are highlighted in red.

**Figure 4 ijms-25-02839-f004:**
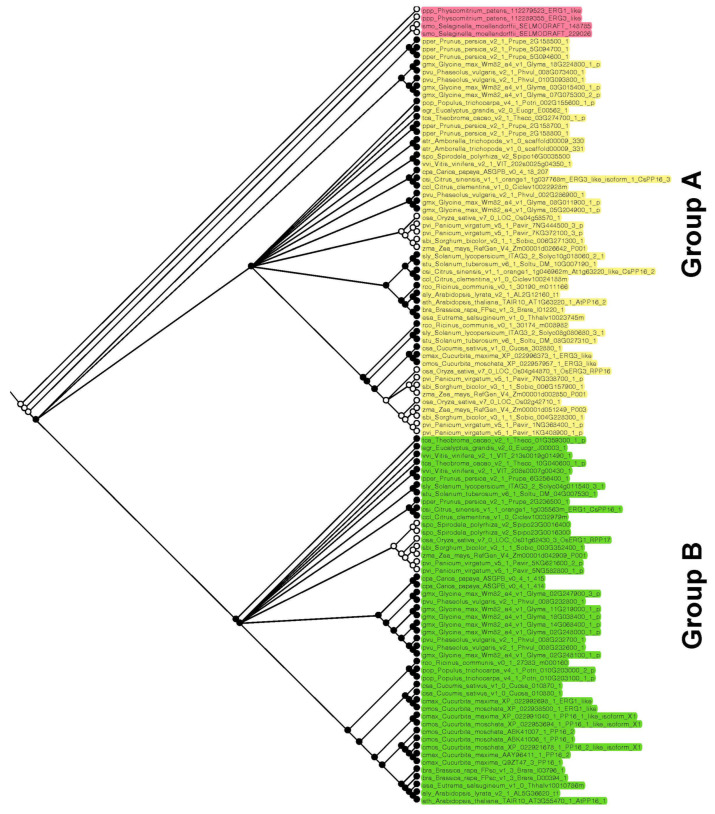
Reconstruction of PP16 ancestral orthologs in Viridiplantae species. The ancestral tree was produced by Bayesian statistics using 104 PP16 ortholog sequences. The analysis of PP16 ancestral orthologs was obtained with the Mesquite software version 3.81.

**Figure 5 ijms-25-02839-f005:**
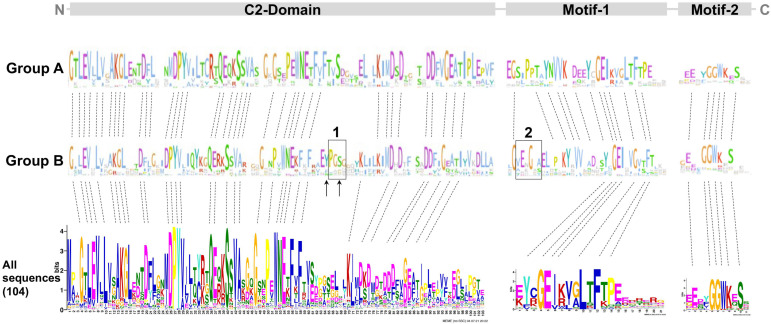
MEME and logo analysis showing common residues and motifs of the group A and B members. C2 comprises most of the protein length. Motifs 1 and 2 are only found in PP16 but not in E-SYT proteins from chlorophytes and embryophytes. The dotted lines indicate conserved amino acid residues. The black rectangles indicate insertion 1 and insertion 2, respectively. The black arrows indicate tyrosine (Y) and serine (S) residues.

**Figure 6 ijms-25-02839-f006:**
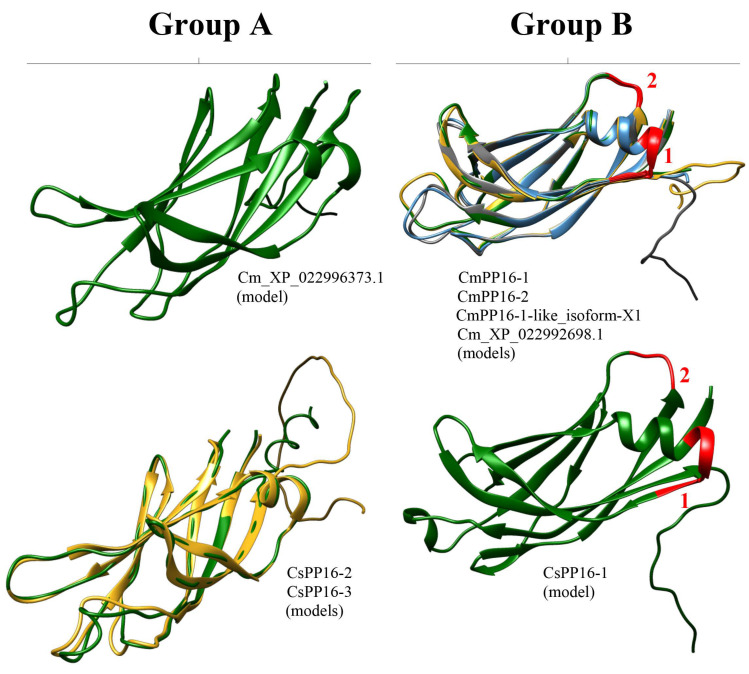
Prediction of PP16 tridimensional structures. Models of representative proteins from A and B groups are shown from *C. sinensis* and *C. maxima*. The red numbers 1 and 2 of the models of the A and B groups indicate insertion 1 and insertion 2, respectively.

**Figure 7 ijms-25-02839-f007:**
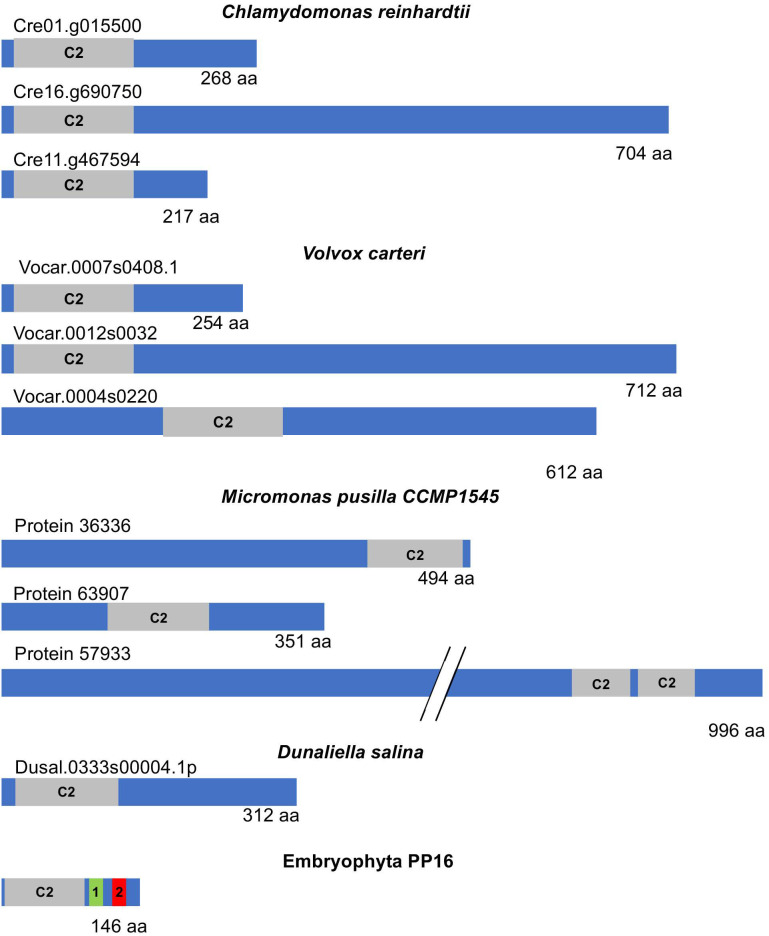
Schematic diagram representing the structure of E-SYT and CmPP16. Extended synaptotagmins (E-SYTs), the closest counterparts of embryophyte PP16 from representative chlorophytes, and CmPP16 as a member of this protein family are shown in the diagrams. The gray boxes indicate the position and size of the C2 domain within the protein. PP16 domains 1 and 2 are highlighted as green and red boxes, respectively. Note the size variation in E-SYT proteins in chlorophytes.

**Figure 8 ijms-25-02839-f008:**
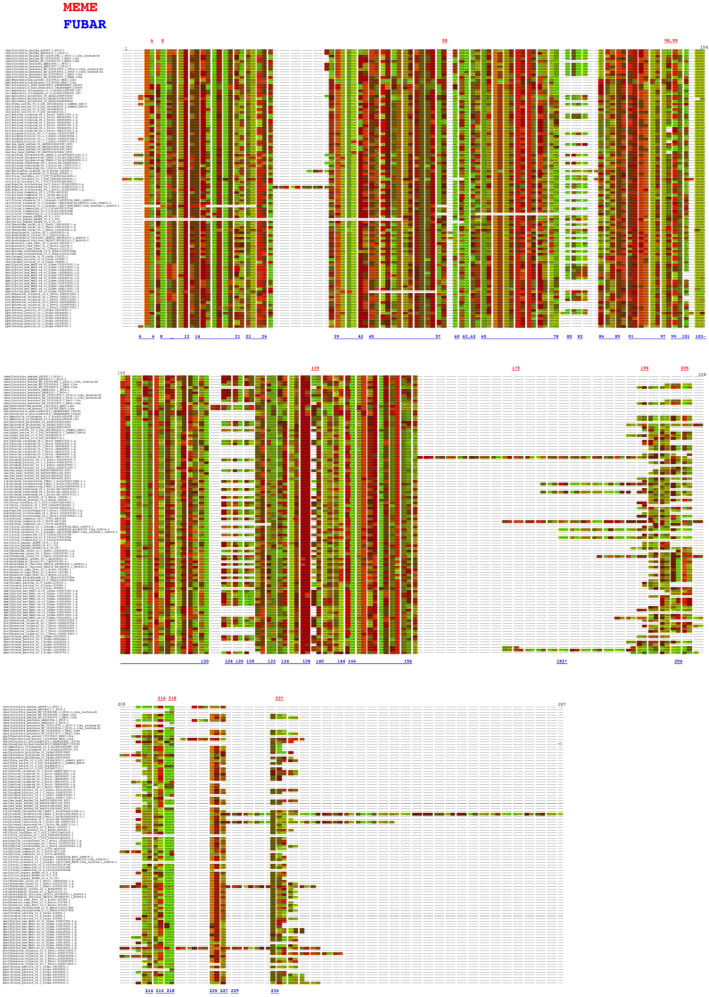
Analysis of PP16 gene selective pressure by the MEME and FUBAR methods. The red and blue color fonts correspond to the MEME and FUBAR methods, respectively.

## Data Availability

Data are contained within the article or Appendix A.

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
