# Peer review of "Evolutionary and Structural Analysis of PP16 in Viridiplantae"

_ijms, 2024, doi:10.3390/ijms25052839_

Round 1

Reviewer 1 Report

Comments and Suggestions for Authors

The paper provides a comprehensive evolutionary and structural analysis of the PP16 gene family in Viridiplantae, highlighting its potential origin and functional diversification.

While the study emphasizes the conservation of the C2 domain across PP16 proteins in vascular plants and their potential roles in RNA binding, intercellular transport, and response to water deficit, the functional implications of these findings in non-vascular plants and chlorophytes remain unclear. Expanding on the functional assays or experimental evidence supporting these roles in a wider array of plant species could significantly strengthen the paper's conclusions.

The authors have utilized SWISS-MODEL, a well-regarded automated protein structure homology-modelling server, to predict the 3D structures of PP16 proteins. While SWISS-MODEL is a valuable tool for such analyses, I recommend the authors also consider using AlphaFold for their structural predictions. The integration of AlphaFold's predictions could complement the results obtained from SWISS-MODEL, potentially offering higher-resolution insights into the structural dynamics and functional domains of PP16 proteins.

Author Response

Reviewer 1

  1. The paper provides a comprehensive evolutionary and structural analysis of the PP16 gene family in Viridiplantae, highlighting its potential origin and functional diversification. While the study emphasizes the conservation of the C2 domain across PP16 proteins in vascular plants and their potential roles in RNA binding, intercellular transport, and response to water deficit, the functional implications of these findings in non-vascular plants and chlorophytes remain unclear. Expanding on the functional assays or experimental evidence supporting these roles in a wider array of plant species could significantly strengthen the paper's conclusions.

Thank you for the observation. We detailed the functions of the PP16 protein in different plant species at the beginning of the discussion. We also mentioned that chlorophytes lack true PP16 proteins, and suggested, based on similarity searches in this taxon, that this protein family derived from synaptotagmins, which are involved in vesicular trafficking, as well as in membrane exchange between organelles. We agree that further work is required to determine the function of the PP16 proteins in more plant species, such as non-vascular plants, but consider that this work is a starting point to determine a more general function for them.

  1. The authors have utilized SWISS-MODEL, a well-regarded automated protein structure homology-modelling server, to predict the 3D structures of PP16 proteins. While SWISS-MODEL is a valuable tool for such analyses, I recommend the authors also consider using AlphaFold for their structural predictions. The integration of AlphaFold's predictions could complement the results obtained from SWISS-MODEL, potentially offering higher-resolution insights into the structural dynamics and functional domains of PP16 proteins.

Thank you for the suggestion. We used AlphaFold (based on the AtPP16-1 template) instead of SWISS-MODEL. Considering your advice, we evaluated the models obtained using both tools and subsequently validated the structures with GalaxyRefine web server. The values obtained for the Ramachandran plots validation were compared and we concluded that the models obtained with AlphaFold achieved better validation scores, so we employed these in our analyses and conclusions. Interestingly, the models generated with AlphaFold were not as similar to the Arabidopsis template, but we discussed the possible reasons.

Reviewer 2 Report

Comments and Suggestions for Authors
  1. Are there any experimental validations planned or underway to confirm the predicted functions of the PP16 protein family, especially in relation to RNA binding and intercellular transport?
  2. Can you provide more details on the methodology used for the in silico structural analysis of the PP16 protein family, especially regarding the identification of disordered domains and their potential roles? 
Comments on the Quality of English Language
  1. Line 40, [1], [2], [3], --> [1,2,3], same for rest to ensure the correct use of in-text citation format.
  2. Line 108-109, font size not consistent with rest in the figure.
  3. Line 149-184, the text in the figure is hard to read clearly.
  4. Line 471-513, the figure's resolution is too low.
  5. Line 639, delete login

Author Response

Reviewer 2

  1. Are there any experimental validations planned or underway to confirm the predicted functions of the PP16 protein family, especially in relation to RNA binding and intercellular transport?

To validate the predicted functions of PP16 in plants, we have improved the discussion section by including several experiments that could be conducted to elucidate the functions of PP16 across different taxa. These experiments include assaying for long-distance movement of PP16 protein and/or mRNA between heterografts, overexpression of this gene in species that clearly lack it (chlorophytes), high-resolution structural analysis experiments, or bioinformatics analyses covering a broader evolutionary spectrum.

  1. Can you provide more details on the methodology used for thein silico structural analysis of the PP16 protein family, especially regarding the identification of disordered domains and their potential roles?

We used the PREDICTER server (http://biomine.cs.vcu.edu/servers/DEPICTER/) for analysis of intrinsically disordered regions of the PP16 models obtained of AlphaFold2 (https://colab.research.google.com/github/sokrypton/ColabFold/blob/main/AlphaFold2.ipynb#scrollTo=33g5IIegij5R). Then, the results of PREDICTER server were compared and discussed with the results previously reported of Sashi et al., 2018.

Regarding the functions of these intrinsically disordered regions, the literature is consistent about the RNA-binding functions of proteins with these regions.

  • Zeke, A., Schad, E., Horvath, T., Abukhairan, R., Szabo, B., & Tantos, A. (2022). Deep structural insights into RNA‐binding disordered protein regions.Wiley Interdisciplinary Reviews: RNA13(5), e1714.
  • Ottoz, D. S., & Berchowitz, L. E. (2020). The role of disorder in RNA binding affinity and specificity.Open Biology10(12), 200328.

  1. Line 40, [1], [2], [3], --> [1,2,3], same for rest to ensure the correct use of in-text citation format.

Already corrected.

  1. Line 108-109, font size not consistent with rest in the figure.

Already corrected.

  1. Line 149-184, the text in the figure is hard to read clearly.

Already corrected.

  1. Line 471-513, the figure's resolution is too low.

Already corrected.

  1. Line 639, delete login

Already deleted.